# Return of Unconditional Generation: A Self-supervised Representation Generation Method

**Tianhong Li**    **Dina Katabi**    **Kaiming He**

MIT CSAIL

## Abstract

Unconditional generation—the problem of modeling data distribution without relying on human-annotated labels—is a long-standing and fundamental challenge in generative models, creating a potential of learning from large-scale unlabeled data. In the literature, the generation quality of an unconditional method has been much worse than that of its conditional counterpart. This gap can be attributed to the lack of semantic information provided by labels. In this work, we show that one can close this gap by generating semantic representations in the representation space produced by a self-supervised encoder. These representations can be used to condition the image generator. This framework, called Representation-Conditioned Generation (RCG), provides an effective solution to the unconditional generation problem without using labels. Through comprehensive experiments, we observe that RCG significantly improves unconditional generation quality: *e.g.*, it achieves a new state-of-the-art FID of 2.15 on ImageNet 256×256, largely reducing the previous best of 5.91 by a relative 64%. Our unconditional results are situated in the same tier as the leading class-conditional ones. We hope these encouraging observations will attract the community's attention to the fundamental problem of unconditional generation. Code is available at https://github.com/LTH14/rcg.

## 1 Introduction

Generative models have been long developed as *unsupervised* learning methods in the history, *e.g.*, in the seminal works including GAN [27], VAE [39], and diffusion models [57]. These fundamental methods focus on learning the probabilistic distributions of data, without relying on the availability of human annotations. This problem, often categorized as *unconditional generation* in today's community, is in pursuit of utilizing the vast abundance of unannotated data to learn complex distributions.

However, unconditional image generation has been largely stagnant in comparison with its conditional counterpart. Recent research [18, 54, 10, 11, 24, 50] has demonstrated compelling image generation quality when conditioned on class labels or text descriptions provided by humans, but its quality degrades *significantly* without these conditions. Closing the gap between unconditional and conditional generation is a challenging and scientifically valuable problem: it is a necessary step towards unleashing the power of large-scale unannotated data, which is a common goal in today's deep learning community.

We hypothesize that such a performance gap is because human-annotated conditioning introduces rich semantic information to simplify the generative process. In this work, we largely close this gap by taking inspiration from a closely related area—unsupervised/self-supervised learning.[1] We

---

[1] In this paper, the term of "unsupervised learning" implies "not using human supervision". Thus, we view self-supervised learning as a form of unsupervised learning. The distinction between these two terminologies is beyond the scope of this work.

38th Conference on Neural Information Processing Systems (NeurIPS 2024).

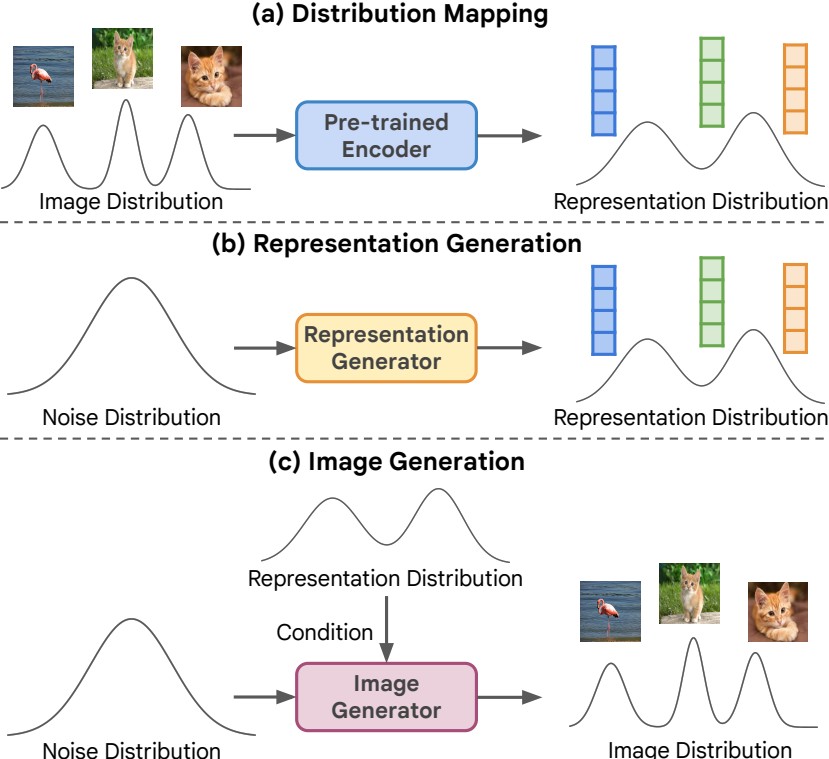

Figure 1: **The Representation-Conditioned Generation (RCG) framework** for unconditional generation. RCG consists of three parts: (a) it uses a pre-trained self-supervised encoder to map the image distribution to a representation distribution; (b) it learns a representation generator that samples from a noise distribution and generates a representation subject to the representation distribution; (c) it learns an image generator (*e.g.*, which can be ADM [18], DiT [50], or MAGE [41]) that maps a noise distribution to the image distribution conditioned on the representation distribution.

observe that the *representations* produced by a strong self-supervised encoder (*e.g.*, [30, 12, 8, 14]) can also capture a lot of semantic attributes without human supervision, as reflected by their transfer learning performance in the literature. These self-supervised representations can serve as a form of conditioning without violating the unsupervised nature of unconditional generation, creating an opportunity to get rid of the heavy reliance on human-annotated labels.

Based on this observation, we propose to first unconditionally generate a self-supervised representation and then condition on this representation to generate the images. As a preprocessing step (Figure 1a), we use a pre-trained self-supervised encoder (*e.g.*, MoCo [14]) to map the image distribution into the corresponding representation distribution. In this representation space, we train a representation generator without any conditioning (Figure 1b). As this space is low-dimensional and compact [65], learning the representation distribution is favorably feasible with unconditional generation. In practice, we implement it as a very lightweight diffusion model. Given this representation space, we train a second generator that is conditioned on these representations and produces images (Figure 1c). This image generator can conceptually be any image generation model. The overall framework, called *Representation-Conditioned Generation* (RCG), provides a new paradigm for unconditional generation.[2]

RCG is conceptually simple, flexible, yet highly effective for unconditional generation. RCG greatly improves unconditional generation quality regardless of the specific choice of the image generator (Figure 2), reducing FID by 71%, 76%, 82%, and 51% for LDM-8, ADM, DiT-XL/2, and MAGE-L, respectively. This indicates that RCG largely reduces the reliance of current generative models on

---

[2]The term "unconditional generation" implies "not conditioned on human labels". As such, RCG is an unconditional generation solution.

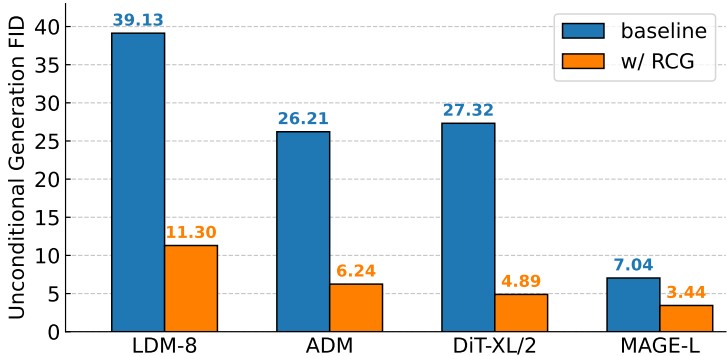

Figure 2: **Unconditional Image Generation can be largely improved by our RCG framework**. Regardless of the specific form of the image generator (LDM [54], ADM [18], DiT [50], or MAGE [41]), RCG massively improves the unconditional generation quality. Generation quality is measured by FID on ImageNet with a 256×256 resolution. All comparisons between models without and with RCG are conducted under controlled conditions to ensure fairness. The technical details and more metrics are in Section 4.1.

manual labels. On the challenging ImageNet 256×256 benchmark, RCG achieves an unprecedented 2.15 FID for unconditional generation. This performance not only largely outperforms previous unconditional methods, but more surprisingly, can catch up with the strong leading methods that are *conditional* on class labels. We hope our method and encouraging results will rekindle the community's interest in the fundamental problem of unconditional generation.

## 2  Related Work

**Generative Models.** Generative models aim at accurately modeling data distribution to generate new data point that resembles the original data. One stream of generative models is built on top of generative adversarial networks (GANs) [27, 69, 37, 70, 7]. Another stream is based on a two-stage scheme [63, 53, 10, 67, 40, 41, 11]: first tokenize the image into a latent space and then apply maximum likelihood estimation and sampling in the token space. Diffusion models [33, 59, 18, 54, 52] have also achieved superior results on image synthesis.

The design of a generative model is mostly orthogonal to how it is conditioned. However, literature has shown that unconditional generation often significantly lags behind conditional generation under the same design[18, 41, 10], especially on complex data distributions.

**Unconditional Generation.** Unconditional generation is the fundamental problem in the realm of generative models. It aims to model the data distribution without relying on human annotations, highlighted by seminal works of GAN [27], VAE [39], and diffusion models [57]. It has demonstrated impressive performance in modeling simple image distributions such as scenes or human faces [23, 10, 18, 54], and has also been successful in applications beyond images where human annotation is challenging or impossible, such as novel molecular design [66, 28, 26], medical image synthesis [71, 16, 47], and audio generation [48, 42, 25]. However, recent research in this domain has been limited, largely due to the notable gap between conditional and unconditional generation capabilities of recent generative models on complex data distributions [46, 18, 19, 41, 3, 61].

Prior efforts to narrow this gap mainly group images into clusters in the representation space and use the cluster indices as underlying class labels to provide conditioning [46, 43, 3, 35]. However, these methods assume that the dataset is clusterable, and the optimal number of clusters is close to the number of classes. Additionally, these methods fall short of generating diverse representations— they are unable to produce different representations within the same cluster or underlying class.

**Representations for Image Generation.** Prior works have explored the effectiveness of exploiting representations for image generation. DALL-E 2 [52], a *text-conditional* image generation model, first converts text prompts into image embeddings, and then uses these embeddings as the conditions

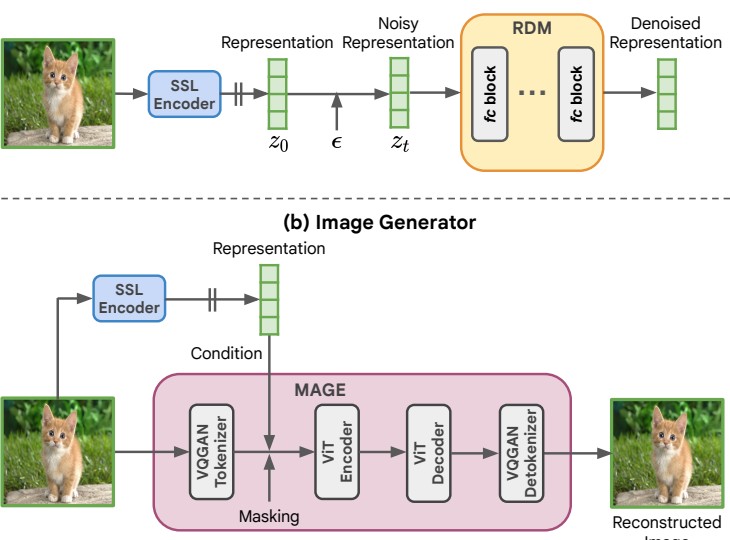

Figure 3: **RCG's training framework.** The pre-trained self-supervised image encoder extracts representations from images and is fixed during training. To train the representation generator, we add standard Gaussian noise to the representations and ask the network to denoise them. To train the MAGE image generator, we add random masking to the tokenized image and ask the network to reconstruct the missing tokens conditioned on the representation extracted from the same image.

to generate images. In contrast, RCG for the first time demonstrates the possibility of directly generating image representations *from scratch*, a necessary step to enable conditioning on self-supervised representations in unconditional image generation. Another work, DiffAE [51], trains an image encoder in an end-to-end manner with a diffusion model as decoder, aiming to learn a meaningful and decodable image representation. However, its semantic representation ability is still limited (e.g., compared to self-supervised models like MoCo [14], DINO [8]), which largely hinders its performance in unconditional generation. Another relevant line of work is retrieval-augmented generative models [5, 4, 9], where images are generated based on representations extracted from existing images. Such semi-parametric methods heavily rely on ground-truth images to provide representations during generation, a requirement that is impractical in many generative applications.

## 3 Method

Directly modeling a complex high-dimensional image distribution is a challenging task. RCG decomposes it into two much simpler sub-tasks: first modeling the distribution of a compact low-dimensional representation, and then modeling the image distribution conditioned on this representation distribution. Figure 1 illustrates the idea. Next, we describe RCG and its extensions in detail.

### 3.1 The RCG Framework

RCG comprises three key components: a pre-trained self-supervised image encoder, a representation generator, and an image generator. Each component's design is elaborated below:

**Distribution Mapping.** RCG employs an off-the-shelf image encoder to convert the image distribution to a representation distribution. This image encoder has been pre-trained using self-supervised contrastive learning methods, such as MoCo v3 [14], on ImageNet. This approach regularizes the representations on a hyper-sphere while achieving state-of-the-art performance in representation learning. The resulting distribution is characterized by two essential properties: it is simple enough to be modeled effectively by an *unconditional* representation generator, and it is rich in high-level semantic content, which is crucial for guiding image generation. These attributes are vital for the effectiveness of the following two components.

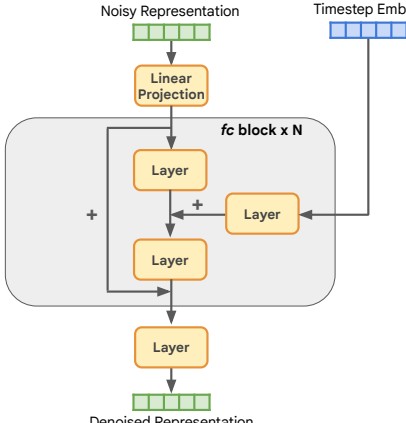

Figure 4: **Representation generator's backbone architecture.** Each "Layer" consists of a LayerNorm layer [1], a SiLU layer [22], and a linear layer. The backbone consists of an input layer that projects the representation to hidden dimension $C$, followed by $N$ fully-connected (fc) blocks, and an output layer that projects the hidden latent back to the original representation dimension. The diffusion timestep is embedded and added to every fc block.

**Representation Generation.** In this stage, we want to generate abstract, unstructured representations without conditions. To address this issue, we develop a diffusion model for unconditional representation generation, which we call a representation diffusion model (RDM). RDM employs a fully-connected network with multiple fully-connected residual blocks as its backbone (Figure 4). Each block consists of an input layer, a timestep embedding projection layer, and an output layer, where each layer consists of a LayerNorm [1], a SiLU [22], and a linear layer. Such an architecture is simply controlled by two parameters: the number of blocks, $N$, and the hidden dimension, $C$.

RDM follows DDIM [58] for training and inference. As shown in Figure 3a, during training, image representation $z_0$ is mixed with standard Gaussian noise variable $\epsilon$: $z_t = \sqrt{\alpha_t}z_0 + \sqrt{1 - \alpha_t}\epsilon$. The RDM backbone is then trained to denoise $z_t$ back to $z_0$. During inference, RDM generates representations from Gaussian noise following the DDIM sampling process [58]. Since RDM operates on highly compacted representations, it brings marginal computation overheads for both training and generation (Appendix B.1), while providing substantial semantic information for the image generator, introduced next.

**Image Generation.** The image generator in RCG crafts images conditioned on self-supervised representations. Conceptually, such an image generator can be any modern conditional image generative model by substituting its original conditioning (e.g., class label or text) with representations. In Figure 3b, we take MAGE [41], a parallel decoding generative model as an example. The image generator is trained to reconstruct the original image from a masked version of the image, conditioned on the representation of the same image. During inference, the image generator generates images from a fully masked image, conditioned on the representation generated by the representation generator.

We experiment with four representative generative models: ADM [18], LDM [54], and DiT [50] are diffusion-based frameworks, and MAGE [41] is a parallel decoding framework. Our experiments show that all four generative models achieve much better performance when conditioned on high-level self-supervised representations (Table 1).

### 3.2 Extensions

Our RCG framework can easily be extended to support guidance even in the absence of labels, and to support conditional generation when desired. We introduce these extensions as follows.

**Enabling Guidance in Unconditional Generation.** In class-conditional generation, the presence of labels allows not only for class conditioning but can also provides additional "guidance" in the generative process. This mechanism is often implemented through classifier-free guidance in class-conditional generation methods [32, 54, 11, 50]. In RCG, the representation-conditioning behavior enables us to also benefit from such guidance, even in the absence of labels.

Specifically, RCG follows [32, 11] to incorporate guidance into its MAGE generator. During training, the MAGE generator is trained with a 10% probability of not being conditioned on image representations, analogous to [32] which has a 10% probability of not being conditioned on labels. For each inference step, the MAGE generator produces a representation-conditioned logit, $l_c$, and

Table 1: **RCG significantly improves the unconditional generation performance of current generative models**, evaluated on ImageNet 256×256. All numbers are reported under the unconditional generation setting.

| Unconditional generation | | FID↓ | IS↑ |
|---|---|---|---|
| LDM-8 [54] | baseline | 39.13 | 22.8 |
| | w/ RCG | **11.30** (−27.83) | **101.9** (+79.1) |
| ADM [18] | baseline | 26.21 | 39.7 |
| | w/ RCG | **6.24** (−19.97) | **136.9** (+97.2) |
| DiT-XL/2 [50] | baseline | 27.32 | 35.9 |
| | w/ RCG | **4.89** (−22.43) | **143.2** (+107.3) |
| MAGE-B [41] | baseline | 8.67 | 94.8 |
| | w/ RCG | **3.98** (−4.69) | **177.8** (+83.0) |
| MAGE-L [41] | baseline | 7.04 | 123.5 |
| | w/ RCG | **3.44** (−3.60) | **186.9** (+63.4) |

an unconditional logit, $l_u$, for each masked token. The final logits, $l_g$, are calculated by adjusting $l_c$ away from $l_u$ by the guidance scale, $\tau$: $l_g = l_c + \tau(l_c - l_u)$. The MAGE generator then uses $l_g$ to sample the remaining masked tokens. Additional implementation details of RCG's guidance are provided in Appendix A.

**Simple Extension to Class-conditional Generation.** RCG seamlessly enables conditional image generation by training a task-specific conditional RDM. Specifically, a class embedding is integrated into each fully-connected block of the RDM, in addition to the timestep embedding. This enables the generation of class-specific representations. The image generator then crafts the image conditioned on the generated representation. As shown in Table 3 and Appendix C, this simple modification allows users to specify the class of the generated image while keeping RCG's superior generative performance, all without the need to retrain the image generator.

## 4 Experiments

We evaluate RCG on the ImageNet 256×256 dataset [17], which is a common benchmark for image generation and is especially challenging for unconditional generation. Unless otherwise specified, we do not use ImageNet labels in any of the experiments. We generate 50K images and report the Frechet Inception Distance (FID) [31] and Inception Score (IS) [55] as the standard metrics for assessing the fidelity and diversity of the generated images. Evaluations of precision and recall are included in Appendix B.1. Unless otherwise specified, we follow the evaluation suite provided by ADM [18]. **All ablations and results on other datasets are included in Appendix B.1.**

### 4.1 Observations

We extensively evaluate the performance of RCG with various image generators and compare it to the results of state-of-the-art unconditional and conditional image generation methods. Several intriguing properties are observed.

**RCG significantly improves the unconditional generation performance of current generative models.** In Table 1, we evaluate the proposed RCG framework using different image generators. The results demonstrate that conditioning on generated representations substantially improves the performance of all image generators in unconditional generation. Specifically, it reduces the FID for unconditional LDM-8, ADM, DiT-XL/2, MAGE-B, and MAGE-L by 71%, 76%, 82%, 54%, and 51%, respectively. We further show that such improvement is also universal across different datasets, as demonstrated by the results on CIFAR-10 and iNaturalist in Appendix B.1. These findings confirm that RCG markedly boosts the performance of current generative models in unconditional generation, significantly reducing their reliance on human-annotated labels.

Moreover, such outstanding performance can be achieved with lower training cost compared to current generative models. In Figure 5, we compare the training cost and unconditional generation FIDs

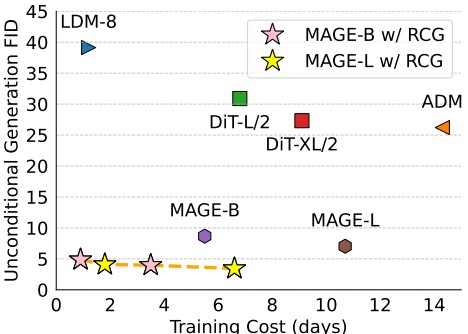

Figure 5: **RCG achieves outstanding unconditional generation performance with less training cost.** All numbers are reported under the unconditional generation setting. The training cost is measured using a cluster of 64 V100 GPUs. Given that the MoCo v3 ViT encoder is pre-trained and not needed for generation, its training cost is excluded. Detailed computational cost is reported in Appendix B.1.

Table 2: **RCG largely improves the state-of-the-art in unconditional image generation** on ImageNet 256×256. All numbers are reported under the unconditional generation setting. Following common practice, we report the number of parameters used during generation. † denotes semi-parametric methods which require ground-truth ImageNet images during generation.

| Unconditional generation | #params | FID↓ | IS↑ |
|---|---|---|---|
| BigGAN [19] | ∼70M | 38.61 | 24.7 |
| ADM [18] | 554M | 26.21 | 39.7 |
| MaskGIT [10] | 227M | 20.72 | 42.1 |
| RCDM† [5] | - | 19.0 | 51.9 |
| IC-GAN† [9] | ∼75M | 15.6 | 59.0 |
| ADDP [61] | 176M | 8.9 | 95.3 |
| MAGE-B [41] | 176M | 8.67 | 94.8 |
| MAGE-L [41] | 439M | 7.04 | 123.5 |
| RDM-IN† [4] | 400M | 5.91 | 158.8 |
| **RCG** (MAGE-B) | 239M | 3.98 | 177.8 |
| **RCG** (MAGE-L) | 502M | 3.44 | 186.9 |
| **RCG-G** (MAGE-B) | 239M | 3.19 | 212.6 |
| **RCG-G** (MAGE-L) | 502M | **2.15** | **253.4** |

of RCG and current generative models. RCG achieves a significantly lower FID with less training cost than current generative models. Specifically, MAGE-B with RCG achieves an unconditional generation FID of 4.87 in less than a day when trained on 64 V100 GPUs. This demonstrates that decomposing the complex tasks of unconditional generation into much simpler sub-tasks can significantly simplify the data modeling process.

**RCG largely improves the state-of-the-art in unconditional image generation.** In Table 2, we compare MAGE with RCG and previous state-of-the-art methods in unconditional image generation. As shown in Figure 8 and Table 2, RCG can generate images with both high fidelity and diversity, achieving an FID of 3.44 and an Inception Score of 186.9. These results are further enhanced with the guided version of RCG (RCG-G), which reaches an FID of 2.15 and an Inception Score of 253.4, significantly surpassing previous methods of unconditional image generation.

**RCG's unconditional generation performance rivals leading methods in class-conditional image generation.** In Table 3, we perform a system-level comparison between the *unconditional* RCG and state-of-the-art *class-conditional* image generation methods. MAGE-L with RCG is comparable to leading class-conditional methods, with and without guidance. These results demonstrate that RCG, for the first time, improves the performance of unconditional image generation on complex data distributions to the same level as that of state-of-the-art class-conditional generation methods, effectively bridging the historical gap between class-conditional and unconditional generation.

In Table 4, we further conduct an apple-to-apple comparison between the class-conditional versions of LDM-8, ADM, and DiT-XL/2 and their unconditional counterparts using RCG. Surprisingly, with RCG, these generative models consistently outperform their class-conditional versions by a noticeable margin. This demonstrates that the rich semantic information from the unconditionally generated representations can guide the generative process even more effectively than class labels.

Table 3: **System-level comparison: RCG's unconditional generation performance rivals leading methods in class-conditional image generation** on ImageNet 256×256. Following common practice, we report the number of parameters used during generation. Class-conditional results are marked in gray.

| Methods | #params | w/o Guidance FID↓ | w/o Guidance IS↑ | w/ Guidance FID↓ | w/ Guidance IS↑ |
|---|---|---|---|---|---|
| *Class-conditional* | | | | | |
| ADM [18] | 554M | 10.94 | 101.0 | 4.59 | 186.7 |
| LDM-4 [54] | 400M | 10.56 | 103.5 | 3.60 | 247.7 |
| U-ViT-H/2-G [2] | 501M | - | - | 2.29 | 263.9 |
| DiT-XL/2 [50] | 675M | 9.62 | 121.5 | 2.27 | 278.2 |
| DiffiT [29] | - | - | - | 1.73 | 276.5 |
| BigGAN-deep [6] | 160M | 6.95 | 198.2 | - | - |
| MaskGIT [10] | 227M | 6.18 | 182.1 | - | - |
| MDTv2-XL/2 [24] | 676M | 5.06 | 155.6 | **1.58** | 314.7 |
| CDM [34] | - | 4.88 | 158.7 | - | - |
| MAGVIT-v2 [68] | 307M | 3.65 | 200.5 | 1.78 | **319.4** |
| RIN [36] | 410M | 3.42 | 182.0 | - | - |
| VDM++ [38] | 2B | **2.40** | **225.3** | 2.12 | 267.7 |
| **RCG, conditional** (MAGE-L) | 512M | 2.99 | 215.5 | 2.25 | 300.7 |
| *Unconditional* | | | | | |
| **RCG** (MAGE-L) | 502M | 3.44 | 186.9 | 2.15 | 253.4 |

Table 4: **Apple-to-apple comparison: RCG's unconditional generation outperforms the class-conditional counterparts of current generative models**, evaluated on ImageNet 256×256. MAGE does not report its class-conditional generation performance. Class-conditional results are marked in gray.

| Methods | | FID↓ | IS↑ |
|---|---|---|---|
| LDM-8 [54] | w/ class labels | 17.41 | 72.9 |
| | **w/ RCG** | **11.30** | **101.9** |
| ADM [18] | w/ class labels | 10.94 | 101.0 |
| | **w/ RCG** | **6.24** | **136.9** |
| DiT-XL/2 [50] | w/ class labels | 9.62 | 121.5 |
| | **w/ RCG** | **4.89** | **143.2** |

As shown in Table 3 and Appendix C, RCG also supports class-conditional generation with a simple extension. Our representation diffusion model can easily adapt to class-conditional representation generation, thereby enabling RCG to also adeptly perform class-conditional image generation. This result demonstrates the effectiveness of RCG in leveraging its superior unconditional generation performance to benefit downstream conditional generation tasks.

Importantly, such an adaptation does not require retraining the representation-conditioned image generator. For any new conditioning, only the lightweight representation generator needs to be retrained. This potentially enables pre-training of the self-supervised encoder and image generator on large-scale unlabeled datasets, and task-specific training of conditional representation generator on a small-scale labeled dataset. We believe that this property, similar to self-supervised learning, allows RCG to both benefit from large unlabeled datasets and adapt to various downstream generative tasks with minimal overheads. We leave the exploration on this direction to future work.

## 4.2 Qualitative Insights

In this section, we showcase the visualization results of RCG, providing insights into its superior generative capabilities. Figure 8 illustrates RCG's unconditional image generation results on ImageNet 256×256. The model is capable of generating both diverse and high-quality images without relying on human annotations. The high-level semantic diversity in RCG's generation stems from

Generated Images

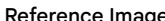
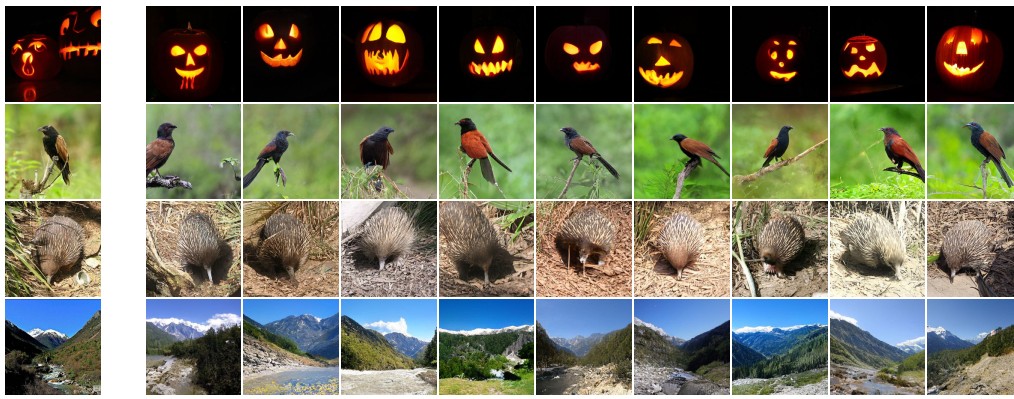

Figure 6: RCG can generate images with diverse appearances but similar semantics from the same representation. We extract representations from reference images and, for each representation, generate a variety of images from different random seeds.

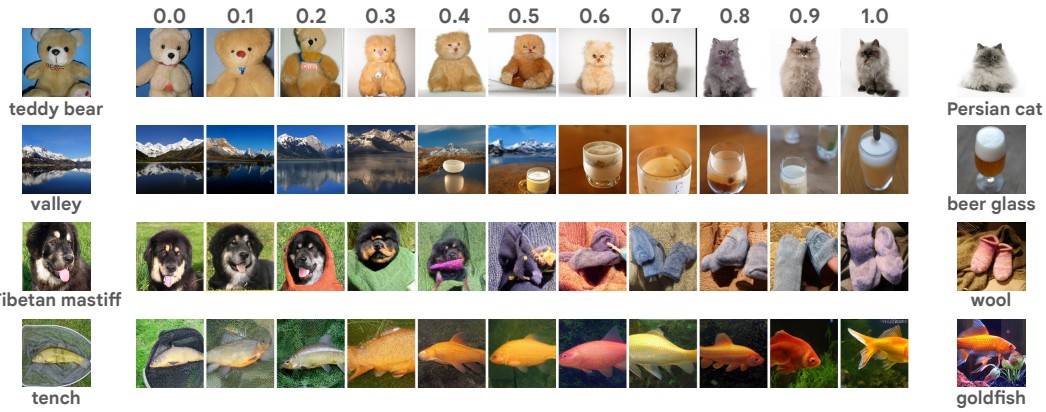

Figure 7: RCG's results conditioned on interpolated representations from two images. The semantics of the generated images gradually transfer between the two images.

its representation generator, which models the distribution of representations and samples them with varied semantics. By conditioning on these representations, the complex data distribution is broken down into simpler, representation-conditioned sub-distributions. This decomposition significantly simplifies the task for the image generator, leading to the production of high-quality images.

Besides high-quality generation, the image generator can also introduce significant low-level diversity in the generative process. Figure 6 illustrates RCG's ability to generate diverse images that semantically align with each other, given the same representation from the reference image. The images generated by RCG can capture the semantic essence of the reference images while differing in specific details. This result highlights RCG's capability to leverage semantic information in representations to guide the generative process, without compromising the diversity that is important in unconditional image generation.

Figure 7 further showcases RCG's semantic interpolation ability, demonstrating that the representation space is semantically smooth. By leveraging RCG's dependency on representations, we can semantically transition between two images by linearly interpolating their respective representations. The interpolated images remain realistic across varying interpolation rates, and their semantic contents smoothly transition from one image to another. For example, interpolating between an image of "Tibetan mastiff" and an image of "wool" could generate a novel image featuring a dog wearing a woolen sweater. This also highlights RCG's potential in manipulating image semantics within a low-dimensional representation space, offering new possibilities to control image generation.

# 5   Discussion

Computer vision has entered a new era where learning from extensive, unlabeled datasets is becoming increasingly common. Despite this trend, the training of image generation models still mostly relies on labeled datasets, which could be attributed to the large performance gap between conditional and unconditional image generation. Our paper addresses this issue by exploring *Representation-Conditioned Generation*, which we propose as a nexus between conditional and unconditional image generation. We demonstrate that the long-standing performance gap can be effectively bridged by generating images conditioned on self-supervised representations and leveraging a representation generator to model and sample from this representation space. We believe this approach has the potential to liberate image generation from the constraints of human annotations, enabling it to fully harness the vast amounts of unlabeled data and even generalize to modalities that are beyond the scope of human annotation capabilities.

**Acknowledgements.**   This work was supported by the GIST MIT Research Collaboration grant funded by GIST. Tianhong Li was also supported by the Mathworks Fellowship. We thank Huiwen Chang, Saining Xie, Zhuang Liu, Xinlei Chen, and Mike Rabbat for their discussion and feedback. We also thank Xinlei Chen for his support on MoCo v3.

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

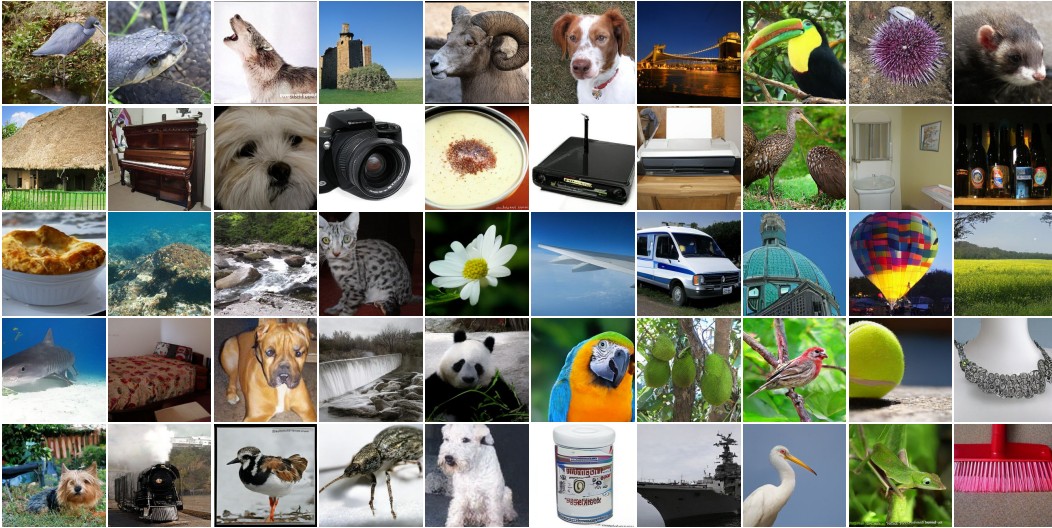

Figure 8: **Unconditional generation results** of RCG on ImageNet 256×256. RCG can generate realistic images with diverse semantics without human annotations.

# A  Implementation Details

In this section, we describe the implementation details of RCG, including hyper-parameters, model architecture, and training paradigm. We also include a copy of our code in the supplementary material. **All codes and pre-trained model weights will be made publicly available.**

**Image Encoder.** We use Vision Transformers (ViTs) [20] pre-trained with MoCo v3 [14] as the default image encoder. We evaluate three ViT variants (ViT-S, ViT-B, and ViT-L) in the main paper, each trained on ImageNet for 300 epochs. We utilize the image representation after the MLP projection head, favoring its adjustable dimensionality. An output dimension of 256 has proven the most effective. The representation of each image is normalized by its own mean and variance. Detailed training recipes of our pre-trained image encoder can be found in [14].

**Representation Diffusion Model (RDM).** Our RDM architecture employs a backbone of multiple fully connected blocks. We use 12 blocks and maintain a consistent hidden dimension of 1536 across the network. The timestep $t$ is discretized into 1000 values, each embedded into a 256-dimensional vector. For class-conditional RDM, we embed each class label into a 512-dimensional vector. Both timestep and class label embeddings are projected to 1536 dimensions using different linear projection layers in each block. Detailed hyper-parameters for RDM's training and generation can be found in Table 5.

**Image Generator.** We experiment with ADM [18], LDM [54], DiT [50], and MAGE [41] as the image generator. For representation-conditioned ADM, LDM and DiT, we substitute the original class embedding conditioning with the image representation. We follow ADM's original training recipe to train representation-conditioned ADM for 400 epochs. We follow LDM-8's original training recipe, with modifications including a batch size of 256, a learning rate of 6.4e-5, and a training duration of 40 epochs. We follow the DiT training scheme in [15], which trains DiT-XL for 400 epochs with batch size 2048 and a linear learning rate warmup for 100 epochs. The $\beta_2$ of the AdamW optimizer is set to 0.95. For representation-conditioned MAGE, we replace the default "fake" class token embedding [C$_0$] with the image representation for conditioning.

During the training of RCG's image generator, the image is resized so that the smaller side is of length 256, and then randomly flipped and cropped to 256×256. The input to the SSL encoder is further resized to 224×224 to be compatible with its positional embedding size. Our implementation of guidance follows Muse [11], incorporating a linear guidance scale scheduling. Detailed hyper-parameters for our representation-conditioned MAGE are provided in Table 6.

Table 5: **RDM implementation details.**

| config | value |
|---|---|
| #blocks | 12 |
| hidden dimension | 1536 |
| #params | 63M |
| optimizer | AdamW [45] |
| learning rate | 5.12e-4 |
| weight decay | 0.01 |
| optimizer momentum | $\beta_1, \beta_2 =$ 0.9, 0.999 |
| batch size | 512 |
| learning rate schedule | constant |
| training epochs | 200 |
| augmentation | Resize(256) RandCrop(256) RandomFlip (0.5) |
| diffusion steps | 1000 |
| noise schedule | linear |
| DDIM steps | 250 |
| $\eta$ | 1.0 |

Table 6: **Repsentation-conditioned MAGE implementation details.**

| config | value |
|---|---|
| optimizer | AdamW [45] |
| base learning rate | 1.5e-4 |
| weight decay | 0.05 |
| optimizer momentum | $\beta_1, \beta_2 = 0.9, 0.95$ |
| batch size | 4096 |
| learning rate schedule | cosine decay [44] |
| warmup epochs | 10 |
| training epochs | 800 |
| gradient clip | 3.0 |
| label smoothing [60] | 0.1 |
| dropout | 0.1 |
| augmentation | Resize(256) RandCrop(256) RandomFlip (0.5) |
| masking ratio min | 0.5 |
| masking ratio max | 1.0 |
| masking ratio mode | 0.75 |
| masking ratio std | 0.25 |
| rep. drop rate | 0.1 |
| parallel-decoding temperature | 6.0 (B) 11.0 (L) |
| parallel-decoding steps | 20 |
| guidance scale ($\tau$) | 1.0 (B) 6.0 (L) |
| guidance scale schedule | linear [11] |

# B   Additional Quantitative Results

## B.1   Ablations

This section provides a comprehensive ablation study of the three core components of RCG. Our default setup uses MoCo v3 ViT-B as the pre-trained image encoder, an RDM with a 12-block, 1536-hidden-dimension backbone trained for 100 epochs, and a MAGE-B image generator trained for 200 epochs. Unless otherwise specified, all other properties and modules are set to the default settings during each component's individual ablation. The FID in this section is evaluated against the ImageNet validation set.

**Distribution Mapping.** Table 7 ablates the image encoder. Table 7a compares image encoders trained via various self-supervised learning methods (MoCo v3, DINO, and iBOT), highlighting

Table 7: **Distribution mapping ablation experiments.** The default encoder is MoCo v3 ViT-B with 256 projection dimension. Default settings are marked in gray.

| Method | FID | IS |
|---|---|---|
| No condition | 14.23 | 57.7 |
| MoCo v3 [14] | 5.07 | 142.5 |
| DINO [8] | 7.53 | 160.8 |
| iBOT [72] | 8.05 | 148.7 |
| DeiT [62] (supervised) | 5.51 | 211.7 |

| Model | params | lin. | FID | IS |
|---|---|---|---|---|
| ViT-S | 22M | 73.2 | 5.77 | 120.8 |
| ViT-B | 86M | 76.7 | 5.07 | 142.5 |
| ViT-L | 304M | 77.6 | 5.06 | 148.2 |

| Projection Dim | FID | IS |
|---|---|---|
| 32 | 9.14 | 81.0 |
| 64 | 6.09 | 119.2 |
| 128 | 5.19 | 143.3 |
| 256 | 5.07 | 142.5 |
| 768 | 6.10 | 112.7 |

(a) **Pre-training.** RCG achieves good performance with encoders pre-trained with different contrastive learning and supervised learning methods.

(b) **Model size.** RCG scales up with larger pre-trained encoders with better linear probing accuracy.

(c) **Projection dimension.** The dimensionality of the image representation is important in RCG's performance.

Table 8: **Representation generation ablation experiments.** The default RDM backbone is of 12 blocks and 1536 hidden dimensions, trained for 100 epochs, and takes 250 sampling steps during generation. The representation Frechet Distance (rep FD) is evaluated between 50K generated representations and representations extracted from the ImageNet training set by MoCo v3 ViT-B. Default settings are marked in gray.

| #Blocks | FID | IS | rep FD |
|---|---|---|---|
| 3 | 7.53 | 113.5 | 0.71 |
| 6 | 5.40 | 132.9 | 0.53 |
| 12 | 5.07 | 142.5 | 0.48 |
| 18 | 5.20 | 141.9 | 0.50 |
| 24 | 5.13 | 141.5 | 0.49 |

| Hidden Dim | FID | IS | rep FD |
|---|---|---|---|
| 256 | 12.99 | 67.3 | 5.98 |
| 512 | 9.07 | 99.8 | 1.19 |
| 1024 | 5.35 | 132.0 | 0.56 |
| 1536 | 5.07 | 142.5 | 0.48 |
| 2048 | 5.09 | 142.8 | 0.48 |

| Epochs | FID | IS | rep FD |
|---|---|---|---|
| 10 | 5.94 | 124.4 | 0.87 |
| 50 | 5.21 | 138.3 | 0.54 |
| 100 | 5.07 | 142.5 | 0.48 |
| 200 | 5.07 | 145.1 | 0.47 |
| 300 | 5.05 | 144.3 | 0.47 |

| #Steps | FID | IS | rep FD |
|---|---|---|---|
| 20 | 5.80 | 120.3 | 0.87 |
| 50 | 5.28 | 133.0 | 0.55 |
| 100 | 5.15 | 138.1 | 0.48 |
| 250 | 5.07 | 142.5 | 0.48 |
| 500 | 5.07 | 142.9 | 0.49 |

(a) **Model depth.** A deeper RDM can improve generation performance.

(b) **Model width.** A wider RDM can improve generation performance.

(c) **Training epochs.** Training RDM longer improves generation performance.

(d) **Diffusion steps.** More sampling steps can improve generation performance.

Table 9: **Image generation ablation experiments.** The default image generator is MAGE-B trained for 200 epochs. Table 9c evaluates different $\tau$ using MAGE-L with RCG trained for 800 epochs and the FID is evaluated following ADM suite. Default settings are marked in gray.

| Conditioning | FID | IS |
|---|---|---|
| No condition | 14.23 | 57.7 |
| Cluster label | 6.60 | 121.9 |
| Class label | 5.83 | 147.3 |
| Generated rep. | 5.07 | 142.5 |
| Oracle rep. | 4.37 | 149.0 |

| Epochs | FID | IS |
|---|---|---|
| 100 | 6.03 | 127.7 |
| 200 | 5.07 | 142.5 |
| 400 | 4.48 | 158.8 |
| 800 | 4.15 | 172.0 |

| $\tau$ | 0.0 | 1.0 | 3.0 | 5.0 | 6.0 | 7.0 |
|---|---|---|---|---|---|---|
| FID | 3.44 | 2.59 | 2.29 | 2.31 | 2.15 | 2.31 |
| IS | 186.9 | 228.5 | 251.3 | 252.7 | 253.4 | 252.6 |

(a) **Conditioning.** Conditioning on generated representations improves over all baselines in FID.

(b) **Training epochs.** Longer training can improve generation performance.

(c) **Classifier-free guidance scale.** $\tau = 6$ achieves the best FID and IS for RCG-L.

their substantial improvements over the unconditional baseline. Additionally, an encoder trained with DeiT [62] in a supervised manner also exhibits impressive performance, indicating RCG's adaptability to both supervised and self-supervised pre-training approaches.

We also notice that using representations from MoCo v3 achieves better FID than using representations from DINO/iBOT. This is likely because only MoCo v3 uses an InfoNCE loss. Literature has shown that optimizing InfoNCE loss can maximize uniformity and preserve maximal information in the representation. The more information in the representation, the more guidance it can provide for the image generator, leading to better and more diverse generation. To demonstrate this, we compute the uniformity loss on representations [65]. Lower uniformity loss indicates higher uniformity and more information in the representation. The uniformity loss of representations from MoCo v3, DINO, and iBOT is -3.94, -3.60, and -3.55, respectively, which aligns well with their generation performance.

Table 7b assesses the impact of model size on the pre-trained encoder. Larger models with better linear probing accuracy consistently enhance generation performance, although a smaller ViT-S model still achieves decent results.

We further analyze the effect of image representation dimensionality, using MoCo v3 ViT-B models trained with different output dimensions from their projection head. Table 7c shows that neither excessively low nor high-dimensional representations are ideal – too low dimensions lose vital image information, while too high dimensions pose challenges for the representation generator.

**Representation Generation.** Table 8 ablates the representation diffusion model and its effectiveness in modeling representation distribution. The RDM's depth and width are controlled by the number of fc blocks and hidden dimensions. Table 8a and Table 8b ablate these parameters, indicating an

Table 10: **CIFAR-10 and iNaturalist results.** RCG consistently improves unconditional image generation performance on different datasets.

| Dataset | Methods | | FID |
|---|---|---|---|
| CIFAR-10 | Improved DDPM [49] | baseline | 3.29 |
| | | **w/ RCG** | **2.62** |
| | | w/ class labels | 2.89 |
| iNaturalist 2021 | MAGE-B | baseline | 8.64 |
| | | **w/ RCG** | **4.49** |
| | | w/ class labels | 4.55 |

optimal balance at 12 blocks and 1536 hidden dimensions. Further, Table 8c and Table 8d suggest that RDM's performance saturates at 200 training epochs and 250 diffusion steps.

Besides evaluating FID and IS on generated images, we also assess the Frechet Distance (FD) [21] between the generated representations and the ground-truth representations. A smaller FD indicates that the distribution of generated representations more closely resembles the ground-truth distribution. Since the MoCo v3 encoder is trained on the ImageNet training set, the representation distribution in the training set can be slightly different from that in the validation set. To establish a better reference point, we compute the FD between 50K randomly sampled representations from the training set and the representations from the entire training set, which should serve as the lower bound of the FD for our representation generator. The result is an FD of 0.38, demonstrating that our representation generator (with an FD of 0.48) can accurately model the representation distribution.

We also evaluate the representation generator against the validation set, resulting in an FD of 2.73. As a reference point, the FD between 50K randomly sampled representations from the training set and the validation set is 2.47, which is also close to the FD of our representation generator.

**Image Generation.** Table 9 ablates RCG's image generator. Table 9a experiments with MAGE-B under different conditioning. MAGE-B with RCG significantly surpasses the unconditional and clustering-based baselines, and further outperforms the class-conditional baseline in FID. This shows that representations could provide rich semantic information to guide the generative process. It is also quite close to the "upper bound" which is conditioned on oracle representations from ImageNet *real* images, demonstrating the effectiveness of the representation generator in producing realistic representations.

We also ablate the training epochs of the image generator and the guidance scale $\tau$, as shown in Table 9b and Table 9c. Training MAGE longer keeps improving the generation performance, and $\tau = 6$ achieves the best FID and IS.

## B.2 Other Datasets

In this section, we evaluate RCG on datasets other than ImageNet to validate its consistent effectiveness across different datasets. We select CIFAR-10 and iNaturalist 2021 [64]. CIFAR-10 represents a relatively simple and low-dimensional image distribution, and iNaturalist 2021 represents a more complex image distribution, with 10,000 classes and 2.7 million images. For CIFAR-10, we employ SimCLR [13] trained on CIFAR-10 as the image encoder and Improved DDPM [49] as the image generator. The FID is evaluated between 50,000 generated images and the CIFAR-10 training set. For iNaturalist, we employ MoCo v3 ViT-B trained on ImageNet as the image encoder and MAGE-B as the image generator. The FID is evaluated between 100,000 generated images and the iNaturalist validation set, which also consists of 100,000 images.

As shown in Table 10, RCG consistently enhances unconditional image generation performance on both CIFAR-10 and iNaturalist 2021, demonstrating its universal effectiveness across various datasets. Notably, the improvement on complex data distributions such as ImageNet and iNaturalist is more significant than on simpler data distributions such as CIFAR-10. This is because RCG decomposes a complex data distribution into two relatively simpler distributions: the representation distribution and the data distribution conditioned on the representation distribution. Such decomposition is particularly effective on complex data distributions, such as natural images, paving the way for generative models to model unlabeled complex data distributions.

Table 11: **Computational cost.** RCG achieves a much smaller FID with similar or less computational cost as baseline methods. The number of parameters, training cost, and the number of training epochs of the representation generator and the image generator are reported separately.

| Unconditional Generation | #Params (M) | Training Cost (days) | Epochs | Throughput (samples/s) | FID |
|---|---|---|---|---|---|
| LDM-8 [54] | 395 | 1.2 | 150 | 0.9 | 39.13 |
| ADM [18] | 554 | 14.3 | 400 | 0.05 | 26.21 |
| DiT-L [50] | 458 | 6.8 | 400 | 0.3 | 30.9 |
| DiT-XL [50] | 675 | 9.1 | 400 | 0.2 | 27.32 |
| MAGE-B [41] | 176 | 5.5 | 1600 | 3.9 | 8.67 |
| MAGE-L [41] | 439 | 10.7 | 1600 | 2.4 | 7.04 |
| **RCG** (MAGE-B) | 63+176 | 0.1+0.8 | 100+200 | 3.6 | 4.87 |
| **RCG** (MAGE-B) | 63+176 | 0.2+3.3 | 200+800 | 3.6 | 3.98 |
| **RCG** (MAGE-L) | 63+439 | 0.3+1.5 | 100+200 | 2.2 | 4.09 |
| **RCG** (MAGE-L) | 63+439 | 0.6+6.0 | 200+800 | 2.2 | 3.44 |

Table 12: RCG's unconditional generation FID, IS, precision and recall on ImageNet 256×256, evaluated following ADM's suite [18].

| Methods | FID↓ | IS↑ | Prec.↑ | Rec.↑ |
|---|---|---|---|---|
| **RCG** (MAGE-B) | 3.98 | 177.8 | 0.84 | 0.47 |
| **RCG** (MAGE-L) | 3.44 | 186.9 | 0.82 | 0.52 |
| **RCG-G** (MAGE-B) | 3.19 | 212.6 | 0.83 | 0.48 |
| **RCG-G** (MAGE-L) | 2.15 | 253.4 | 0.81 | 0.53 |

## B.3 Computational Cost

In Table 11, we present a detailed evaluation of RCG's computational cost, including the number of parameters, training costs, and generation throughput. The training cost of all image generators is measured using a cluster of 64 V100 GPUs. The training cost of RDM is measured using 1 V100 GPU, divided by 64. The generation throughput is measured on a single V100 GPU. As LDM and ADM measure their generation throughput on a single NVIDIA A100 [54], we convert it to V100 throughput by assuming a ×2.2 speedup of A100 vs V100 [56].

As shown in the Table 11, RCG requires significantly lower training costs to achieve great performance. For instance, it achieves an FID of 4.87 in less than one day of training. Moreover, the training and inference costs of the representation generator are marginal compared to those of the image generator. This efficiency potentially enables for lightweight adaptation to various downstream generative tasks by training only the representation generator on small-scale labeled datasets.

## B.4 Precision and Recall

In Table 12, we report the unconditional generation precision and recall of RCG, evaluated on ImageNet 256×256 following the ADM suite [18]. Larger models as well as incorporating guidance (RCG-G) both improve recall while slightly decreases precision.

## C Additional Qualitative Results

We include more qualitative results, including class-unconditional image generation (Figure 9), class-conditional image generation (Figure 10 and Figure 11), and the comparison between generation results with or without guidance (Figure 13). All these results demonstrate RCG's superior performance in image generation. We also include some failure cases in Figure 12.

## D Limitations and Negative Impact

**Limitations.** Like any other generative models, RCG can also produce unrealistic or low-quality results (see Appendix C for some examples).

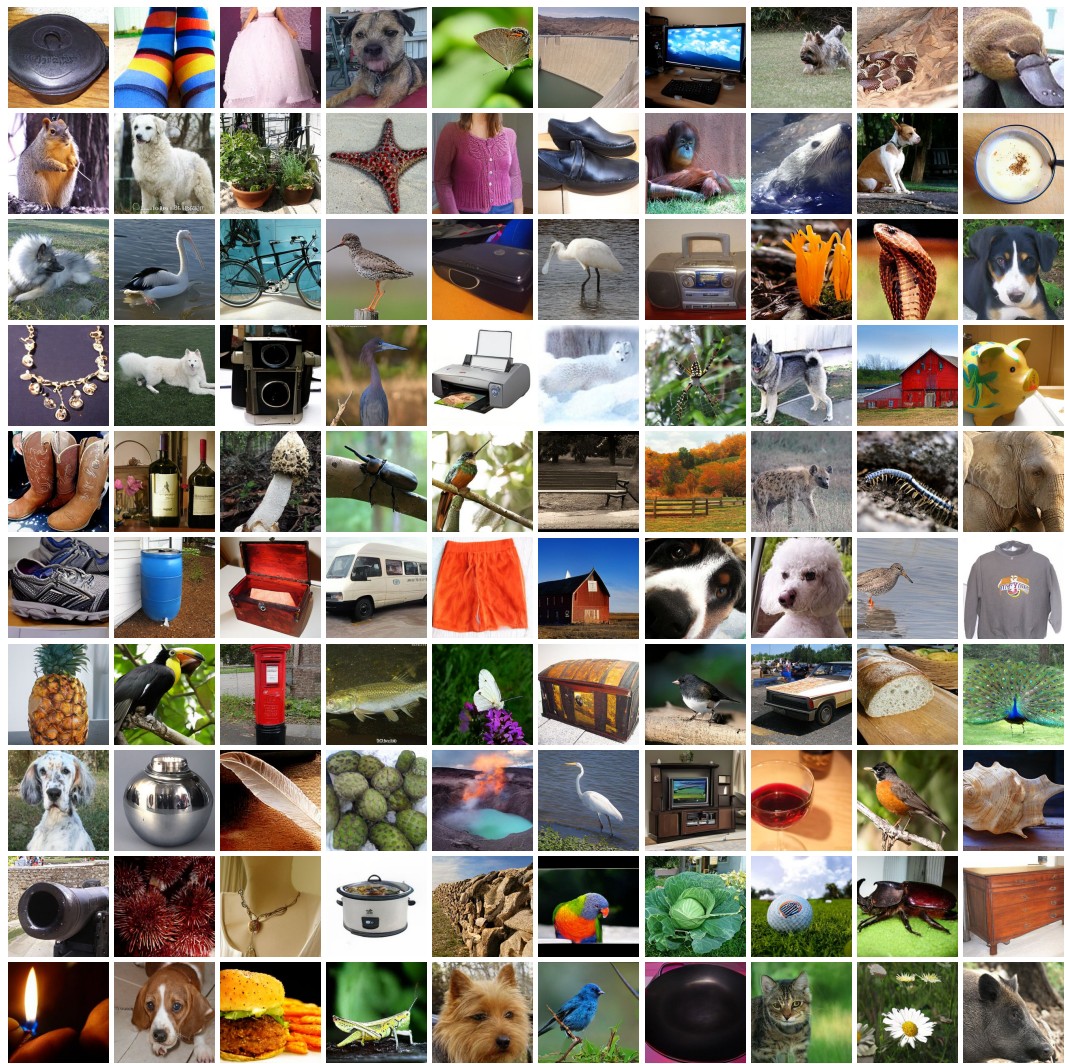

Figure 9: More RCG class-unconditional image generation results on ImageNet 256×256.

**Societal Impact.** Despite the rapid advancements in generative models, they also carry potential negative societal impacts. For instance, such models can amplify existing biases present in internet data. RCG, being a generative model, is not immune to these issues. However, it is important to note that RCG operates within an unconditional generation framework, which does not depend on human-provided labels. This characteristic might possess the potential to mitigate the influence of human biases, offering a more neutral approach to data generation compared to traditional conditional models.

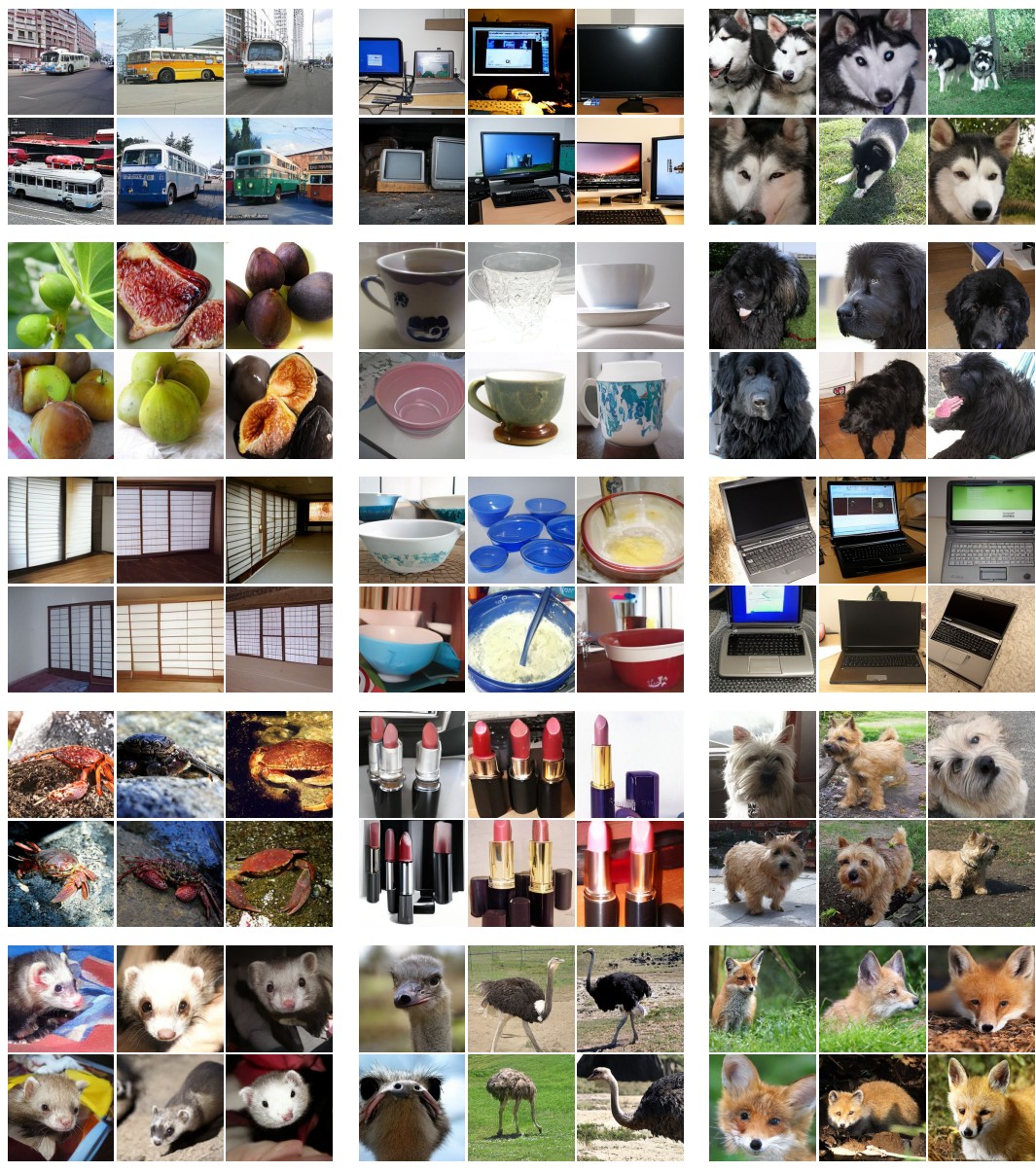

Figure 10: RCG class-conditional image generation results on ImageNet 256×256. Classes are 874: trolleybus, 664: monitor, 249: malamute; 952: fig, 968: cup, 256: Newfoundland; 789: shoji, 659: mixing bowl, 681: notebook; 119: rock crab, 629: lipstick, 192: cairn; 359: ferret, 9: ostrich, 277: red fox.

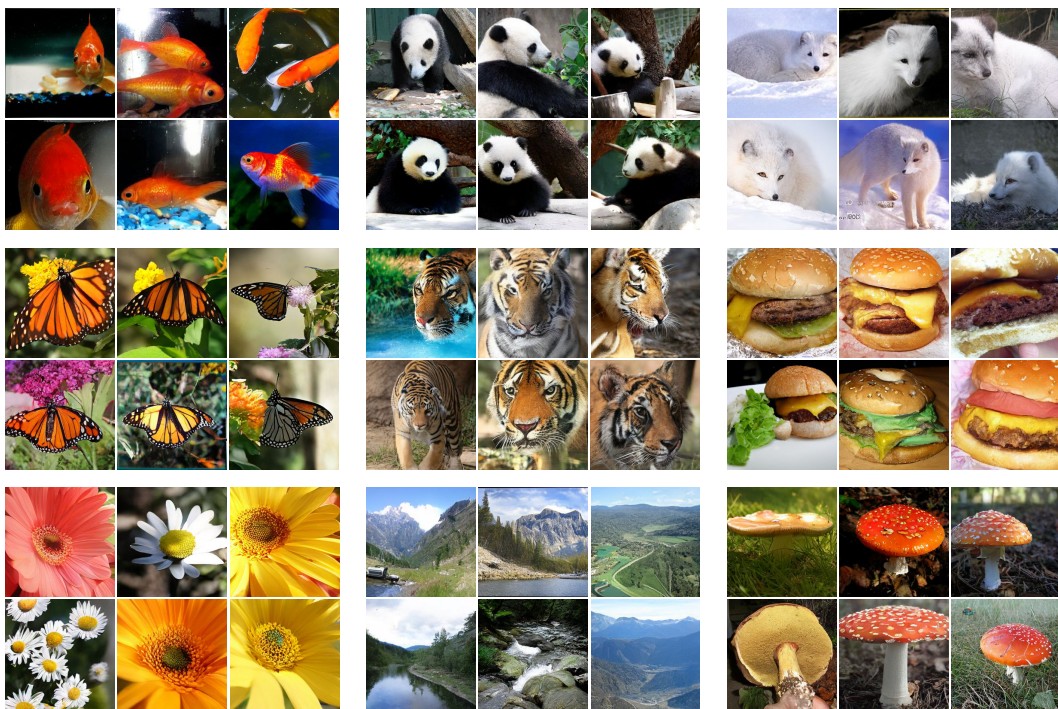

Figure 11: RCG class-conditional image generation results on ImageNet 256×256. Classes are 1: goldfish, 388: panda, 279: Arctic fox; 323: monarch butterfly, 292: tiger, 933: cheeseburger; 985: daisy, 979: valley, 992: agaric

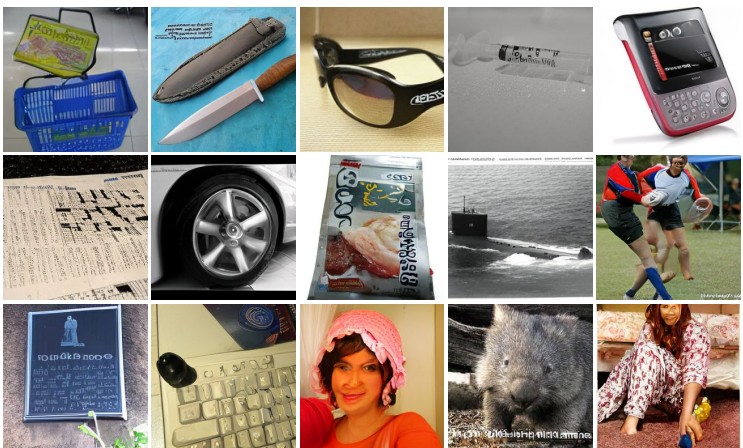

Figure 12: Similar to other generative models on ImageNet, RCG also could face difficulty in generating texts, regular shapes (such as keyboard and wheel), and realistic human.

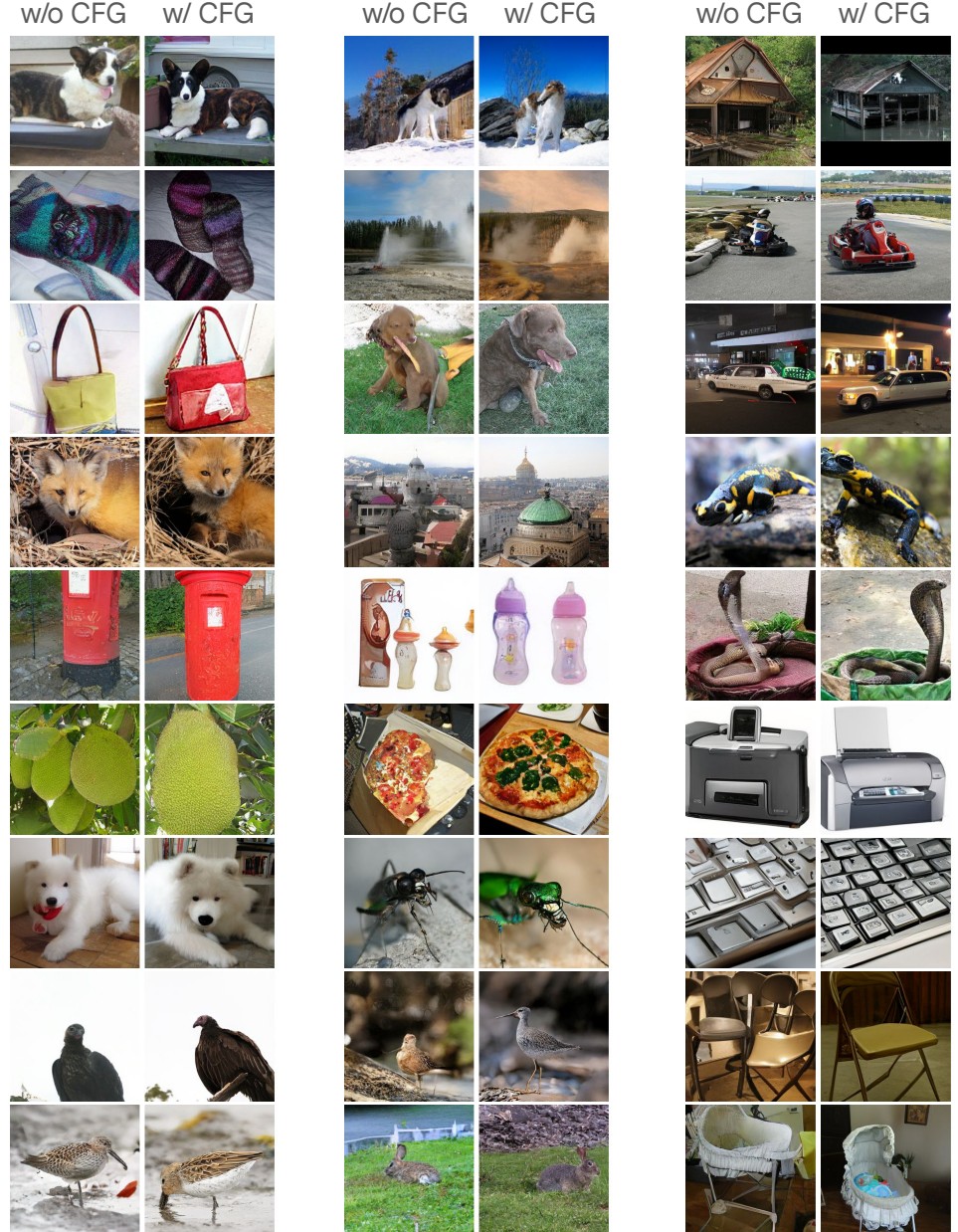

Figure 13: Class-unconditional image generation results on ImageNet 256×256, with or without guidance. RCG achieves strong generation performance even without guidance. Incorporating guidance further improves the generation quality.

