# OpenReview forum: "Return of Unconditional Generation: A Self-supervised Representation Generation Method"
_NeurIPS.cc/2024/Conference — NeurIPS 2024 oral_

### Official Review · Reviewer_pEoh · 2024-07-05

**Soundness:** 4
**Presentation:** 3
**Contribution:** 2
**Rating:** 8
**Confidence:** 4

**Summary:**

The paper proposes a framework, coined Representation-Conditioned Generation (RCG), that aims to bring the advantages of Conditional Generation techniques to Unconditional Generation settings.
To do so, they use the output of a representation generator network instead of class labels.
This representation generator is trained in a prior stage to approximate the distribution of image features extracted by a pre-trained self-supervised network (MoCo-v3).
They test this technique with a set of very different generative models, including a latent diffusion model, a diffusion transformer, and a masked generative transformer, showing improvement accross the board on the ImageNet Unconditional Generation benchmark.

**Strengths:**

a. The proposed method is sound, the technical details are delivered clearly and thoroughly.

b. Experiments and ablations are extensive, and results look robust, being confirmed for very different classes of generative models.

c. The paper provides new empirical evidence and numbers on very relevant points:
  - Diffusion models can generate self-supervised representations convincingly.
  - Those generated features can serve as good conditioning signals to help train image generation models.

**Weaknesses:**

a. Technical novelty is limited: [48] also generates images by first generating pretrained features to condition the image generator, and works such as [5,A] manage to generate high-quality images using representations. The remaining differentiator to existing work in terms of technical contribution is then in the choice of the pre-trained feature extractor and image generator.

b. In that light, while thorough ablation results are presented, I find the interpretation of the results lacking. If those choices are important, and it seems to be the case considering the range of FID scores in the ablations, it would be very useful to provide intuitions about how to make those choices and why.

--

[A] SODA: Bottleneck Diffusion Models for Representation Learning. Hudson et al. CVPR 2024

**Questions:**

- In Table 8, it is difficult to have an interpretation of the FD as we don't have a reference point. Maybe showing FD between training and validation set could be an option, provided the number of samples for the estimates is handled carefully. Also, FD is evaluated on the training set. It would be good to have validation scores as well.

- In a similar vein, I'd like to see the performance (FID) of RCG on features sampled from training images (similar to Figure 6). This would give another indication of how good the representation generation is. This is important since the representation generation is a major, and arguably the most important, contribution of the paper.

- The authors mention in Appendix B that the FID in the corresponding section are computed on the validation set. Is that also the case for the other values in the main paper?

- For the point raised in weakness b., a precise question I'd like to ask is why, as seen in Table 7a, MoCo v3 features are so much better for this task than DINO or iBOT features that are supposed to be better in other downstream tasks.

- Also linked to weakness b., there are strong connections between generative models and self-supervised learning. In fact, works such as [61, A] or [41] on which the current paper builds a lot of their results are already generative models that explicitly double as representation learners, or even have representation learning as their main objective. Those ties should be discussed in the paper.
Moreover, if a generative model can be a representation learner, and if using a representation can help achieve a better generative model, how to explain the good results in the submission becomes even less clear. I'd be happy to hear the authors' thoughts on that point.

In any case, I think the submission is already sound and of interest. It shows very good and solid results on a relevant topic, and opens up interesting research directions. But also, it doesn't provide much in terms of technical novelty or in terms of analysis. The paper touches upon, but barely scratches the surface of very fundamental questions related to self-supervised methods and the training of generative models. I would definitely consider raising my rating to 7 or 8 if it would have a slightly more complete evaluation of the feature generation part and if it provided a deeper analysis of the results.

**Limitations:**

The authors make an attempt to address limitations by showing failed generation. They could discuss the scope in terms of data type: are there results likely to transfer to data? what hurdles to expect in doing so?

They also try to adress societal impacts by discussing generative model biases and hypothesize that unsupervised models should significantly mitigate the influence of human bias. I'd appreciate if they would either expand on that point or alternatively refrain from making such an assumption without a stronger backing. ImageNet photos certainly have been produced by humans and are not bias-free.
Since they are willing to discuss the limitations and impacts of generative models in general, they could also mention potential misuses such as deep fakes and misinformation.

---

> ### Author Rebuttal · Authors · 2024-08-06
>
> We thank the reviewer for appreciating our method, extensive and robust experiments, and the potential to open up interesting research directions. Below we address the weaknesses (***W***) and questions (***Q***) raised by the reviewer. We hope the reviewer could consider raising the score given the additional evaluation in feature generation and the interpretation of the ablation results provided in the rebuttal.
>
> ***Technical novelty (W1)***
>
> As emphasized in the general response, one major technical contribution of this work is the representation generator. Unlike previous works such as [5, A] ([48] not related to image generation) which require ground-truth images to provide the representation during the generative process, our approach does not rely on such impractical requirements and demonstrates the possibility of unconditionally generating pre-trained self-supervised representations.
>
> ***Additional evaluation of feature generation (Q1, Q2)***
>
> As per the reviewer’s request, below we provide the reference points for FD and RCG’s performance using representations sampled from training images.
>
> ***Q1: Reference points for FD***
>
> Since the MoCo v3 encoder is trained on the ImageNet training set, the representation distribution in the training set can be slightly different from that in the validation set. To establish a better reference point, we compute the FD between 50K randomly sampled representations from the training set and the representations from the entire training set, which should serve as the lower bound of the FD for our representation generator. The result is an FD of 0.38, demonstrating that our representation generator (with an FD of 0.48) can accurately model the representation distribution.
>
> We also evaluate the representation generator against the validation set, resulting in an FD of 2.73. As a reference point, the FD between 50K randomly sampled representations from the training set and the validation set is 2.47, which is also close to the FD of our representation generator. We will include both results in the revision.
>
> ***Q2: Performance using representations sampled from training images***
>
> In Table 9(a), we evaluate our image generator under different conditions, including oracle representations from ImageNet training images. The oracle conditioning yields 4.37 FID and 149.0 IS, while conditioning on our generated representations achieves 5.07 FID and 142.5 IS. This further demonstrates the effectiveness of our representation generator in producing realistic and high-quality representations.
>
> ***FID evaluation scheme (Q3)***
>
> For FID in the main paper, we follow ADM’s evaluation suite which computes FID w.r.t. the ImageNet training set. This evaluation suite is widely adopted in prior works, so we need to follow it to make a fair comparison with them. Evaluating on the training set versus the validation set does not significantly change FID, and the ablation results trend remains consistent. To ensure consistency with the main paper, we will revise the FID results in the ablation section accordingly.
>
> ***Interpretation of the ablation results (W2)***
>
> Most of the results in Tables 7-9 are standard hyper-parameter sweeps. Through these experiments, we aim to provide insights into which hyper-parameters are important and require tuning for future applications of our system.
>
> Two results are particularly interesting: Table 9(a), which we have interpreted in our response to ***Q2***, and Table 7(a), which ablates different pre-trained encoders. We will explain the findings from Table 7(a) below.
>
> ***MoCo v3 features vs. DINO/iBOT features (W2, Q4)***
>
> In Table 7(a), using representations from MoCo v3 achieves better FID than using representations from DINO/iBOT. This is likely because only MoCo v3 uses an InfoNCE loss. Literature has shown that optimizing InfoNCE loss can maximize uniformity and preserve maximal information in the representation [1]. The more information in the representation, the more guidance it can provide for the image generator, leading to better and more diverse generation. To demonstrate this, we compute the uniformity loss on representations following section 4.1.2 from [1]. Lower uniformity loss indicates higher uniformity and more information in the representation. The uniformity loss of representations from MoCo v3, DINO, and iBOT is -3.94, -3.60, and -3.55, respectively, which aligns well with their generation performance. We will include this result and discussion in the revision.
>
> [1] Understanding Contrastive Representation Learning through Alignment and Uniformity on the Hypersphere
>
> ***Generative models and self-supervised learning (W2, Q5)***
>
> We thank the reviewer for initiating this discussion. The community has a long-standing belief in the synergy between self-supervised learning and generative models: good representations should enhance the generative process, and generative models can learn robust representations. As the reviewer mentioned, several papers have provided evidence supporting the latter. On the other hand, we focus on the former, showing that explicitly providing generated representations can significantly improve unconditional generative models, which also offers new and compelling evidence to support the synergy.
>
> ***Limitations and negative societal impacts***
>
> Applying RCG to other data types is beyond this paper's scope and could be an interesting future direction. It requires a pre-trained encoder, typically available off-the-shelf for common data types. We will include this discussion in the revision.
>
> We acknowledge that image datasets can contain various types of human bias, including biases in data collection and labeling. RCG's unsupervised nature could mitigate labeling bias as it does not rely on human-provided labels. However, we agree that this topic is beyond the paper’s scope and will refrain from making this claim in the revision. We will also include discussions on potential misuses.

---

> ### Comment · Reviewer_pEoh · 2024-08-12
>
> I thank the authors for the detailed responses, I have no major concerns left.
> After going through the paper, all the reviews, and the rebuttals, I believe it is a strong submission that contains significant results and prompts important discussions. As such, I would recommend acceptance and I updated my rating accordingly.

---

### Official Review · Reviewer_fB14 · 2024-07-09

**Soundness:** 2
**Presentation:** 3
**Contribution:** 1
**Rating:** 5
**Confidence:** 5

**Summary:**

This paper proposes generative models conditioned on representation obtained from a pre-trained self-supervised encoder to achieve high-quality diverse generation.

**Strengths:**

1. The writing is clear.
2. The experimental results demonstrate significant improvement over unconditional generation.

**Weaknesses:**

1. The idea of generation conditioned on representation is not new. As the authors mentioned, conditioning on instance representations has been well studied in the community. Though the authors argue that they require ground-truth images to provide representations during generation, the issue can be addressed by adopting the idea from VAE to align those instance representations with the prior noise or by incorporating the representation generator proposed in this paper. Therefore, I would say the main contribution of this work is to improves those representation generation works by introducing a generative modeling for the representation, which however is limited.
2. The deep clustering community has also adopted the technique of self-supervised learning for a more clustering friendly representation and has achieved significant improvement in complex datasets, like ImageNet.  Therefore, generation conditioned on clustering structures also has potential to bring gaps between unconditional generation and conditional generation. I would say the studied problem has been well addressed in the community. Comparisons with conditioning on clustering should be included.

**Questions:**

1. When extending to class-conditional generation, is it required to fine-tune the model?

**Limitations:**

The limitations should include the discussions on the necessity of a self-supervised encoder and how to obtain such an encoder for datasets, especially for other modalities such as text, speech.

---

> ### Author Rebuttal · Authors · 2024-08-06
>
> We thank the reviewer for appreciating our clear writing and strong experiment results. Below we address the weaknesses (***W***) and questions (***Q***) raised by the reviewer.
>
> ***Importance of the studied problem***
>
> We respectfully disagree with the reviewer that “the studied problem has been well addressed in the community.” Unconditional image generation has lagged behind its conditional counterpart for a long time in the literature, especially on complex data distributions such as ImageNet. As shown in Table 2, none of the prior works achieve an unconditional generation FID less than 5 on ImageNet 256x256, and the previous state-of-the-art (RDM-IN) requires the ImageNet training set during generation to achieve a 5.91 FID. In contrast, as shown in Table 3, state-of-the-art class-conditional methods can easily achieve an FID around or below 2, demonstrating a significant gap between unconditional and class-conditional generation.
>
> Other reviewers have also acknowledged the difficulty and significance of the studied problem. Reviewer xRbx noted, “unconditional image generation has remained stagnant compared to conditional generation.” Reviewer zDRz stated, “the proposed method is designed to solve difficult problems very simply and intuitively.” Reviewer pEoh mentioned, “the paper shows very good and solid results on a relevant topic, touches upon the surface of very fundamental questions related to self-supervised methods and the training of generative models.”
>
> Our paper proposes a novel method based on generating self-supervised representations to address this long-standing open problem in the community. The proposed RCG framework significantly improves the quality of unconditional generation, regardless of the specific form of the image generator. It achieves an unprecedented unconditional generation FID of **2.15**, bridging the long-standing performance gap between unconditional and class-conditional generation methods for the first time. We hope the reviewer could reconsider the importance of the studied problem and the rating of our paper in light of this clarification.
>
> ***Conditioning on instance representations (W1)***
>
> Prior methods for unconditional generation that use instance representations require ***existing images*** to provide representations during generation, which is impractical for many generative applications. Moreover, none of the prior works use a generative model to accurately model the pre-trained self-supervised representation distribution. Our RCG framework is the first unconditional generation framework that generates pre-trained self-supervised representations ***from scratch*** and uses them as conditions for the image generator. This novel approach significantly boosts unconditional generation performance and rivals class-conditional generation methods, all without the need for any images during the generation process.
>
> ***Conditioning on clustering (W2)***
>
> We agree that using pseudo-labels from clustering methods as class labels can be an option for unconditional generation. In fact, we have included this ablation study in Table 9(a), where we experimented with our image generator under different conditions, including clustering labels obtained from MoCo v3 representations. Conditioning on clustering labels achieves 6.60 FID and 121.9 IS, while conditioning on ground-truth class labels achieves 5.83 FID and 147.3 IS. Conditioning on our generated representations achieves 5.07 FID and 142.5 IS. These results demonstrate that clustering-based conditions perform worse than ground-truth class labels, whereas our generated representations can outperform ground-truth class labels. This is because the generated representations provide richer semantic information to guide the generative process. Furthermore, common clustering methods require the dataset to exhibit clear and distinct groupings or clusters that can be easily identified by clustering algorithms. It also limits the diversity of conditions, as it cannot produce different conditions within the same cluster. We will include this discussion in the revision.
>
> ***Extending to class-conditional generation (Q1)***
>
> RCG seamlessly enables class-conditional image generation and achieves competitive performance, as shown in Table 3 (RCG, conditional (MAGE-L)). This is accomplished by training a class-conditional representation generator without the need to retrain or fine-tune the image generator. As shown in Table 11, training the RDM is very lightweight compared to training the image generator.
>
> ***Limitations***
>
> We thank the reviewer for the suggestion. Applying RCG to other data types is beyond this paper's scope and could be an interesting future direction. It requires a pre-trained encoder to extract representations from the data, which are typically available off-the-shelf for common data types such as images, videos, texts, and speech [1, 2, 3, 4, 5, 6]. We will include this discussion in the revision.
>
> [1] An Empirical Study of Training Self-Supervised Vision Transformers
>
> [2] Spatiotemporal Contrastive Video Representation Learning
>
> [3] A Large-Scale Study on Unsupervised Spatiotemporal Representation Learning
>
> [4] SimCSE: Simple Contrastive Learning of Sentence Embeddings
>
> [5] W2v-bert: Combining contrastive learning and masked language modeling for self-supervised speech pre-training
>
> [6] Speech simclr: Combining contrastive and reconstruction objective for self-supervised speech representation learning

---

> > ### Comment · Reviewer_fB14 · 2024-08-11
> > **Thanks for the responses.**
> >
> > I thank the authors point out their ablation study about conditioning on cluster labels. So, I raised my score.
> >
> > However, I still feel that the technical contribution of this paper is marginal as I demonstrated in Weakness 1. Moreover, as shown in their results, the performance of conditioning on cluster labels is relatively good, though a little worse than the proposed method. That proves that generation conditioned on clustering structures has great potential to bring gaps between unconditional generation and conditional generation. Image generation without labels does not lag so behind its conditional counterpart with labels.

---

### Official Review · Reviewer_xRbx · 2024-07-13

**Soundness:** 4
**Presentation:** 3
**Contribution:** 4
**Rating:** 8
**Confidence:** 4

**Summary:**

This paper proposes RCG, a novel framework to enhance unconditional image generation by leveraging self-supervised representations. The main idea is first to generate self-supervised representations unconditionally using a pre-trained encoder and then condition the image generation on these representations. This process does not require human-annotated labels and involves training a lightweight diffusion model to generate representations efficiently. The authors demonstrate the effectiveness of RCG by significantly improving the quality of unconditional generation across various architectures, closing the performance gap between unconditional and conditional generation methods.

**Strengths:**

**[S1]** The paper is well-motivated, addressing the importance of unconditional image generation for utilizing abundant data, which has remained stagnant compared to conditional generation.

**[S2]** The idea of utilizing self-supervised learning for image generation makes sense and is a novel idea.

**[S3]** The paper shows significant performance improvements.

**Weaknesses:**

I found no weaknesses.

**Questions:**

**[Q]** Is there any intuition as to why MoCo-v3 is the best representation for RCG?

**Limitations:**

They addressed the limitations.

---

> ### Author Rebuttal · Authors · 2024-08-06
>
> We thank the reviewer for appreciating our motivation, novel idea, and significant performance improvements. Below we address the question raised by the reviewer.
>
> ***Why MoCo-v3 is the best representation for RCG***
>
> We compare self-supervised representations from different methods in Table 7(a), and the representations from MoCo v3 achieve the best FID. This is likely because MoCo v3 uses an InfoNCE loss, which attracts positive samples and repels negative samples. Literature has shown that optimizing such an InfoNCE loss can maximize uniformity and preserve maximal information in the representation [1], thus providing substantial guidance for the image generator and leading to better and more diverse generation results.
>
> Nonetheless, Table 7(a) also demonstrates that RCG achieves substantial improvements over the unconditional baseline using representations from various image encoders, including self-supervised encoders such as iBOT and DINO, as well as the supervised encoder DeiT. This shows that RCG can effectively utilize different self-supervised encoders and consistently improve the performance of unconditional generation.
>
> [1] Understanding Contrastive Representation Learning through Alignment and Uniformity on the Hypersphere

---

> > ### Comment · Reviewer_xRbx · 2024-08-11
> >
> > Thank you for your response! After reading the other reviews and the authors' comments, I stick to my score and recommend acceptance.

---

### Official Review · Reviewer_zDRz · 2024-07-15

**Soundness:** 4
**Presentation:** 4
**Contribution:** 4
**Rating:** 9
**Confidence:** 5

**Summary:**

This paper proposes a very simple unsupervised image generation framework that does not rely on human labeled annotation without compromising generation quality. This framework has two stages: i) representation generator learning and ii) image generator learning. The representation generator is trained in the form of diffusion model training to take a noisy latent image as input and output the corresponding representation encoded by a self-supervised image representation model. The image generator is trained to generate the image corresponding the given representation (encoded by the self-supervised representation model). This paper demonstrates that this simple framework is effective in achieving generation quality comparable to the counterpart supervised learning methods, regardless of architecture type.

**Strengths:**

1. Simple but effective framework for unsupervised image generation. The proposed method is designed to solve difficult problems very simply and intuitively, so it will be able to provide inspiration not only in this field but also in a variety of other fields.

2. Great presentation. It was very helpful in understanding this paper since it explained the information covered in each section in a very informative but concise manner.

3. A variety of experiments can support the authors' claim and the effectiveness of the proposed method.

**Weaknesses:**

I do not have a major concern for this work. I only have two questions in the method.

1. When training the image generator, why was the representation of the SSL model used instead of the output of the representation generator trained in the previous step? Although the representation generator is trained to distill the generation capability of the SSL model, isn't it more appropriate to train under the same conditions as in the inference in which the output (representation) of the representation generator is used?

2. Is there any way for the two generators to be trained together in an end-to-end manner?

**Questions:**

Please respond the two questions I raised in the weakness section.

**Limitations:**

This paper properly deals with the limitation and potential societal impact in the supplementary material.

---

> ### Author Rebuttal · Authors · 2024-08-06
>
> We thank the reviewer for appreciating our simple but effective framework, the strong experiment results, and the great presentation of our paper. Below, we address the two questions (***Q***) raised by the reviewer.
>
> ***Training the image generator using ground-truth representations vs. generated representations (Q1)***
>
> The reviewer asks why we use the representations from the SSL model instead of the output of the representation generator during the training of the image generator. This is because the image generator needs both the representation as a condition and the corresponding ground-truth image as supervision during training. For a representation output by the unconditional representation generator, it is challenging to determine the corresponding image. Therefore, if we were to use generated representations as conditions for our image generator, we wouldn’t have the corresponding ground-truth images needed for training. Additionally, our design allows us to fully decouple the training of the representation generator and the image generator, making our framework more flexible to train.
>
> ***End-to-end training (Q2)***
>
> This is an excellent suggestion. Training the representation generator and the image generator together could potentially further enhance the performance of the entire system. However, similar to ***Q1***, one possible issue is that the denoised representation output from the representation generator might not match the ground-truth representation and the corresponding ground-truth image, especially at high noise levels. In this scenario, using the denoised representation as the condition for the image generator while using the ground-truth image for supervision might cause inconsistencies during training. Nonetheless, we believe this would be an interesting future direction to explore.

---

> > ### Comment · Reviewer_zDRz · 2024-08-13
> >
> > I appreciate authors to address my questions. I will increase my rating.

---

> > ### Public Comment · ~Philip_Bachman1 · 2024-11-22
> > **Latent variables with non-collapsing posteriors**
> >
> > Just think of the representation space of the MoCo encoder as a latent space and the diffusion model trained over this space as a prior over the latents. You can induce a reasonable posterior distribution over a finite sample of these latents using the softmax formed by scaled cos sim between the MoCo representation for an input image X and a finite sample of representations generated by the diffusion-based prior over MoCo representations. You can sample a generated representation based on that softmax and use it to condition generation of X. You could also play around with, eg, varying the temperature of the softmax in order to limit the information capacity of the latent variable (higher softmax entropy means less info about X). We can think of the softmax in contrastive learning, when evaluated only over negative samples, as estimating a conditional distribution over a non-parametric dictionary of representations that assigns higher probability to representations which are most similar to the representation of the "positive sample". This conditional should work reasonably well for sampling "semantic" information about X to use in conditioning. This also avoids a train/test mismatch in the representations used for conditioning the decoder, since they're always samples from the prior.
> >
> > Approaches like the one described above decouple decisions about what information to cram in a latent variable and how to decode that information to produce a generated image. In the olden days, with deep VAEs and such, there were often issues with "posterior collapse" when adding a more powerful decoder p(x|z). Basically, z would be ignored since the model for p(x|z) was powerful enough to just act like p(x) without much impact on the training likelihoods. In the "representation conditioned" setting described above, we can decide what information should be in the latent variable (eg, whatever MoCo happens to capture), and how much information to condition on when generating X. In the setup described above, the amount of info about X is limited to the log of the number of prior samples in the contrastive softmax minus the entropy of the contrastive softmax. This lets us define a nice latent space and non-collapsing posteriors over that latent space which represent strictly controlled amounts of "semantic" information which can guide data generation.
> >
> > I also think this setup would work fine without the diffusion model over MoCo representations. You could probably get away with just defining the prior as uniform over the appropriate hypersphere. This would make training and sampling a bit quicker. It's also straightforward to extend this approach to hierarchical latents, latents with variable bandwidth, etc.

---

### Author Rebuttal · Authors · 2024-08-06

We thank all reviewers for providing lots of insightful and constructive feedback. We will definitely improve our manuscript accordingly. We are glad to see the commonly recognized strengths highlighted by the reviewers:

1. The presentation of the paper is clear and concise (zDRz, fB14, pEoh).

2. The problem studied in the paper is difficult (zDRz), of importance (xRbx), and relevant (pEoh).

3. The introduced framework is novel (xRbx), sound (pEoh), and intuitive (zDRz). It “will be able to provide inspiration not only in unconditional image generation but also in a variety of other fields” (zDRz), “opening up interesting research directions” (pEoh).

4. The empirical results are extensive and robust (zDRz, pEoh). The performance improvement on unconditional generation is significant (xRbx, fB14).

We would like to reemphasize a major technical contribution of this work: we demonstrate the possibility of ***unconditionally generating a representation*** pre-trained by state-of-the-art self-supervised learning methods. These generated representations can be used as conditions to improve the unconditional generation performance of various image generators. Such a representation generator is key to enabling unconditional generation without relying on ground-truth images during the generative process.

Furthermore, we note that the contribution of our work extends beyond the technical aspects. The ground-breaking empirical finding that unconditional generation can rival the performance of conditional generation by generating and conditioning on representations is a significant contribution. We believe this approach and the promising results have the potential to liberate image generation from the constraints of human annotations and rekindle the community’s interest in the fundamental problem of unconditional generation.

As there are no outstanding common questions, we will address each reviewer’s specific questions in separate responses. We are also happy to continue the discussion if the reviewers have any further questions or concerns.

---

### Decision · Program_Chairs · 2024-09-25

**Decision:**

Accept (oral)

**Comment:**

This paper addresses the problem of unconditional image generation using latent representations. An image generator is conditioned on pre-computed image representations, at the generation stage. The idea, though very simple seems to result in high generation quality surpassing several existing baselines. All the reviewers voted for acceptance and I concur with their decision.